# First Report of Plasmid-Mediated Macrolide-Clindamycin-Tetracycline Resistance in a High Virulent Isolate of *Cutibacterium acnes* ST115

**DOI:** 10.3390/pathogens12111286

**Published:** 2023-10-27

**Authors:** Md Shohel Rana, Jungmin Kim, Shukho Kim

**Affiliations:** 1Department of Biomedical Sciences, The Graduate School, Kyungpook National University, Daegu 41944, Republic of Korea; msranabau5@gmail.com (M.S.R.); minkim@knu.ac.kr (J.K.); 2Department of Microbiology, School of Medicine, Kyungpook National University, Daegu 41944, Republic of Korea

**Keywords:** *Cutibacterium acnes*, antibiotic resistance, plasmid, *erm*(50)

## Abstract

*Cutibacterium acnes*, a prevalent skin commensal, has emerged as a significant global challenge due to its widespread antibiotic resistance. To investigate the antibiotic resistance mechanisms and clinical characterization of *C. acnes* in Korea, we collected 22 clinical isolates from diverse patient specimens obtained from the National Culture Collection for Pathogens across Korea. Among the isolates, KB112 isolate was subjected to whole genome sequencing due to high resistance against clindamycin, erythromycin, tetracycline, doxycycline, and minocycline. The whole genome analysis of KB112 isolate revealed a circular chromosome of 2,534,481 base pair with an average G + C content of 60.2% with sequence type (ST) 115, harboring the potential virulent CAMP factor pore-forming toxin 2 (CAMP2), the multidrug resistance ABC transporter ATP-binding protein YknY, and the multidrug efflux protein YfmO. The genomic sequence also showed the existence of a plasmid (30,947 bp) containing the *erm*(50) and *tet*(W) gene, which confer resistance to macrolide–clindamycin and tetracycline, respectively. This study reports plasmid-mediated multi-drug resistance of *C. acnes* for the first time in Korea.

## 1. Introduction

*Cutibacterium acnes*, a lipophilic, anaerobic, Gram-positive bacterium, is an important commensal living on human skin and an etiological agent of human acne vulgaris, sarcoidosis, joint prosthesis infections, prostate cancer, endocarditis, and osteomyelitis of the humerus [1]. As *C. acnes* is known to be involved in the pathological processes of multiple diseases and infections, antibiotics such as tetracyclines, oral macrolides, and topical clindamycin have been used for decades [2]. Nevertheless, antibiotic resistance reports are steadily increasing because of the extensive and inappropriate use of antibiotics. In general, *C. acnes* exhibits a high prevalence of resistance to tetracycline, doxycycline, clindamycin, erythromycin, and minocycline [3]. Moreover, antibiotic resistance and the hemolytic pattern of *C. acnes* strains are phylogroup-dependent, and there is a correlation between their molecular phylogroups and hemolytic patterns. Wright et al. reported the correlation between hemolytic phenotype and clindamycin resistance [4], and *C. acnes* isolates of phylotypes IA and IB were more likely to be hemolytic than phylotype II isolates [5]. However, the resistance mechanisms against erythromycin and clindamycin in *C. acnes* involve mutations in 23S rRNA and methylation of 23S rRNA through the ribosomal methylase gene *erm*(X) [6]. Macrolide and clindamycin-resistant *C. acnes* strains harboring 23S rRNA mutations are the prevalent group of resistant strains, emerging from antibiotic exposure in individuals with a history of antibiotic use [7]. Recently, the highest rate of erythromycin (26.7%) and clindamycin (30%) resistance in *C. acnes* was observed in Korea [8]. The molecular surveillance of antibiotic resistance and related mechanisms has significantly improved due to the extensive application of complete genome sequencing. Nonetheless, the number of genome-wide analyses of antibiotic-resistant *C. acnes* strains is limited, with only a few published reports rather than in-depth genetic studies. However, the microbiological characteristics of *C. acnes* with a plasmid have not been previously reported in Korea; this is necessary to understand the plasmid-mediated antibiotic-resistant mechanisms, clinical settings, and epidemiological risks. To determine the prevalence and molecular mechanisms underlying plasmid-mediated resistance in *C. acnes*, we present the first molecular analysis of *C. acnes*, an ST115 isolate harboring a plasmid with *erm*(50) and *tet*(W) genes in Korea.

## 2. Materials and Methods

### 2.1. Bacterial Culture Conditions

*C. acnes* were cultured under anaerobic conditions on blood agar plates containing 5% sheep blood or in BHI (Brain Heart Infusion, Becton-Dickinson, Sparks, MD, USA) broth for 3–4 days using the GasPak EZ Anaerobic Pouch System (Becton-Dickinson, Sparks, MD, USA) or an anaerobic chamber (Bactron, Cornelius, NC, USA) connected to an anaerobic gas mixture composed of 90% N_2_, 5% CO_2_, 5% H_2_. Hemolytic activity was observed on blood agar plates (BAP).

### 2.2. Genomic DNA Preparation and Genome Sequencing

Genomic DNA was isolated from *C. acnes* KB112 using the TruSeq Nano DNA Kit (Macrogen, Inc., Seoul, Republic of Korea) following the manufacturer’s guidelines. The integrity of the extracted genomic DNA was assessed by electrophoresis on a 1% agarose gel. Genome sequencing was performed by Macrogen (Macrogen, Inc., Seoul, Republic of Korea) using the Illumina HiSeq platform (Illumina, Inc., San Diego, CA, USA). Data were analyzed using the de novo assembly SPAdes 3.13.0. The genome sequence of *C. acnes* HKGB4 (GenBank RefSeq assembly accession no. GCF_021496585.1) was used as a reference for the assembly.

### 2.3. Sequence Analysis

The translated coding DNA sequences (CDSs) were searched against the nonredundant database of the National Center for Biotechnology Information (NCBI). Additional analysis was performed for gene prediction and functional annotation using the RAST (Rapid Annotation using Subsystem Technology) server database [9]. The presence of protein-coding genes was determined using the NCBI prokaryotic genome annotation pipeline. The genomic feature map of *C. acnes* KB112 and the plasmid pKB112 was generated using the Proksee web service (https://proksee.ca/, accessed on 3 March 2023) and annotated using PROKKA [10].

### 2.4. Linearization of Plasmid DNA

The plasmid was extracted from *C. acnes* KB112 using the GeneAll^®^ Exgene™ Cell SV mini plasmid extraction kit (GeneAll Biotechnology Co, Seoul, Republic of Korea) with lysozyme, following the manufacturer’s guidelines. The plasmid DNA was linearized using the endonuclease *Age*I (New England Biolabs, Inc., Ipswich, MA, USA) and assessed by electrophoresis on a 1% agarose gel.

### 2.5. Phylogenetic and Genomic Arrangement Analysis

Phylogenetic analyses were conducted to determine the relationship between *C. acnes* KB112 and similar bacterial strains using the 16S rRNA sequence from the sequenced genome of *C. acnes* KB112. Pairwise sequence comparisons were performed using BLAST (https://blast.ncbi.nlm.nih.gov (accessed on 3 March 2023)) to identify the nearest relatives of *C. acnes* KB112. A phylogenetic tree was created using the Mega11 tool [11]. The genomic arrangements of *C. acnes* KB112 were investigated using the progressiveMauve algorithm (v2.3.1) [12].

## 3. Results and Discussion

The draft genome sequence of *C. acnes* KB112 has been deposited in GenBank with the accession no. JAODIJ00000000. The genome of *C. acnes* KB112 is a circular chromosome of 2,534,481 bp with an average G + C content of 60.2% (Figure 1A). There are 2374 protein-coding genes, 52 RNA genes (rRNA 3, tRNA 45, and ncRNA 4), and 72 pseudogenes among the 2498 predicted genes. The majority of predicted CDSs (2446 genes) were classified as putative proteins, and the remaining ones were characterized as hypothetical proteins. The genome sequence of isolate KB112 revealed the presence of genes associated with inflammation, virulence, and antimicrobial resistance. The ABC transporters (YknY and YfmO) protein-coding genes are expressed in KB112, which are crucial for cell viability, virulence, pathogenicity, and resistance to antimicrobials. These transporters facilitate efflux functions, enabling the expulsion of pathogenesis-related proteins, hydrolytic enzymes, toxins, and antibiotics [13]. However, the *yfmo* efflux gene found in KB112 is likely associated with resistance to macrolide-lincosamide-streptogramin B antibiotics [14]. Additionally, this strain may express several hydrolases proteins, which are putative host-interacting factors and probably involved in its inflammatory response, as observed in acne vulgaris [15]. 

The most significant finding in our studies was the identification of a circular plasmid (GenBank accession no. OQ053204), denoted pKB112, with a length of 30,947 bp, and a G + C content of 65.5%. An asymmetric nucleotide composition was observed near the minimum cumulative skew point of the plasmid of KB112 (Figure 1B). This asymmetry might correspond to the origin of replication, and putative replication initiates bidirectionally from the end of the genome sequence [16]. The plasmid contained type IV secretory system conjugative DNA transfer family protein, which may be associated with conjugative transfer (Table 1) [17]. Moreover, RAST analysis revealed the presence of two antibiotic resistance genes in plasmid pKB112 (Figure 1C), of which the tetracycline-resistant gene *tet*(W) showed 100% sequence identity with *tet*(W) from *C. granulosum* (GenBank accession no. AP026711) and *C. acnes* TP-CU389 plasmid pTZC1 (GenBank accession no. LC473083). Tet(W) confers resistance to tetracycline using ribosomal protection mechanisms like Tet(M) and Tet(O) [18]. Another gene was supposed to be a member of the erm family, as it shared 100% nucleotide and amino acid similarities with the 23S rRNA adenine N-6-methyltransferase of plasmid pTZC1 from *C. acnes* TP-CU389. This gene of the erm family was denoted *erm*(50) and was located on the *Tn552* transposase, which exhibits a high level of macrolide–clindamycin resistance through mutational changes or enzymatic modification of the antibiotic target. In contrast, *erm*(X) is located on the transposon *Tn5432*, facilitating its horizontal transfer between *C. acnes* strains [17]. 

Additionally, to confirm the identity and integrity of the plasmid DNA from *C. acnes* KB112, the circular plasmid DNA was linearized using the endonuclease *Age*I, resulting four distinct DNA fragments (13,792, 8118, 5135, and 3817 bp, respectively) (Appendix A). This enzymatic cleavage pattern confirmed the presence and approximate sizes of the expected fragments within the plasmid DNA, providing valuable insights into its structure and verifying its identity, as reported in plasmid pTZC1 [17].

Hence, to understand the microbiologic characteristics obtained through genetic analysis to gain insight into the infection mechanisms, we identified the existence of CAMP factor pore-forming toxin 2 (CAMP2) in the genome of *C. acnes* Kb112. CAMP2 is involved in hemolytic activity, which is commonly utilized by bacterial pathogens to break down tissues, invade host cells, disseminate, and counteract the host’s immune defenses [13]. In *C. acnes*, the presence of CAMP2 is considered a virulence factor as it can trigger inflammation. This factor is predominantly expressed by phylotype IA strains, which are associated with hemolysis in *C. acnes* and serve as a molecular marker of virulence [19]. To observe hemolytic activity, *C. acnes* KB112 and *C. acnes* ATCC 11,828 were anaerobically cultured on blood agar plates containing 5% sheep blood. *C. acnes* KB112 exhibited β-hemolysis (clearing of blood agar), whereas *C. acnes* ATCC 11,828 did not exhibit hemolytic activity (Appendix A). The β-hemolysis pattern of *C. acnes* is phylogroup-dependent and correlates with both virulence and antibiotic resistance. Our previous study demonstrated that *C. acnes* KB112 belongs to phylotype 1A_1_ and sequence type (ST) 115, exhibited hemolytic activity, and was highly resistant to tetracycline, doxycycline, clindamycin, erythromycin, and minocycline [3].

The phylogenetic analysis revealed that *C. acnes* KB112 showed a high degree of similarity to *C. acnes* DSM 1897 (100%), and *C. acnes* TP-CU389 (99.93%) (Appendix A). Interestingly, *C. acnes* TP-CU389 harbors a plasmid pTZC1 (31440 bp), which transferred this plasmid to *C. acnes* TP-CU426 (accession no. AP026713) and *C. granulosum* TP-CG7 (accession no. AP026711) with a 500 bp deletion of nucleotide. The plasmid pKB112 showed 100% (30947/30947) identity with the acquired plasmid pTZC1 of *C. acnes* TP-CU426 and *C. granulosum* TP-CG7 [20]. Therefore, the genomic arrangements of *C. acnes* KB112 and *C. acnes* TP-CU389 (accession no. AP019664) are shown in Figure 2. Each genome is displayed horizontally, and homologous portions are demonstrated as colored blocks between the genomes. The blocks that are shifted below in *C. acnes* KB112 represent reversed segments of the *C. acnes* TPCU389 genome. No homologous regions are present outside of these blocks, and the height of the similarity profile represents the average degree of sequence conservation. Although the genomic arrangement similarity between these two genomes is relatively low, the ability of *C. acnes* TPCU389 to transfer plasmids between strains and *cutibacterium* species suggests that it can also transfer plasmids with different STs of *C. acnes*. Hence, it is predicted that the plasmid pKB112 may have originated from the Japanese *C. acnes* TPCU389 strain and was subsequently transferred to Korea.

In conclusion, this study uncovers significant genomic characteristics of the *C. acnes* ST115 KB112 strain and its association with infections and antibiotic resistance. The identification of plasmid-mediated multi-drug resistant *C. acnes* for the first time in Korea will contribute to the surveillance and monitoring of *C. acnes* infections and highlights the importance of genomic analysis in controlling antibiotic-resistant strains to ensure effective treatment strategies.

## Figures and Tables

**Figure 1 pathogens-12-01286-f001:**
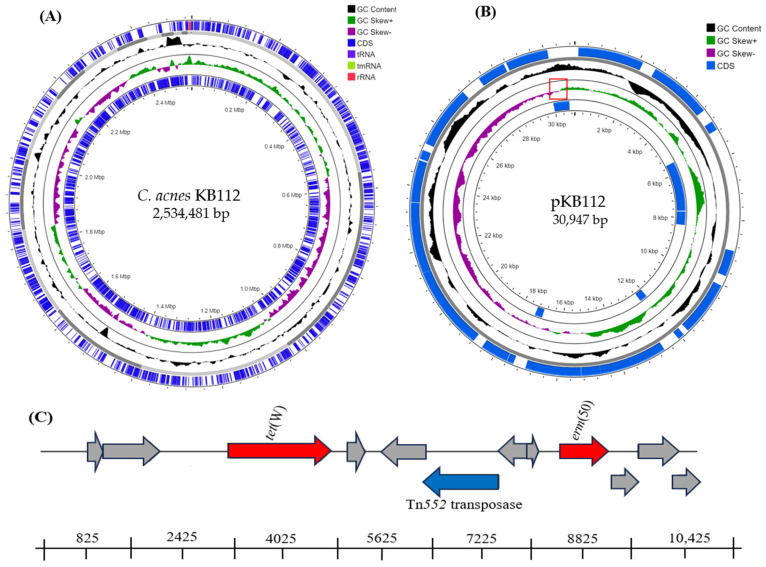
Graphical representation of (**A**) *C. acenes* KB112 chromosome and (**B**) plasmid pKB112. Marked characteristics, including CDS on the forward and reverse strands, tRNA, rRNA, G + C content, and GC skew, are displayed from the edge to the center. The red pane in the plasmid pKB112 represents the putative origin of replication. (**C**) Schematic diagram of the resistance genes and neighboring genes in pKB112; red arrows represent the resistance genes of tetracycline and macrolide-clindamycin respectively, blue arrows represent the genes related to Tn552 transposase, and all gray arrows represent hypothetical genes.

**Figure 2 pathogens-12-01286-f002:**
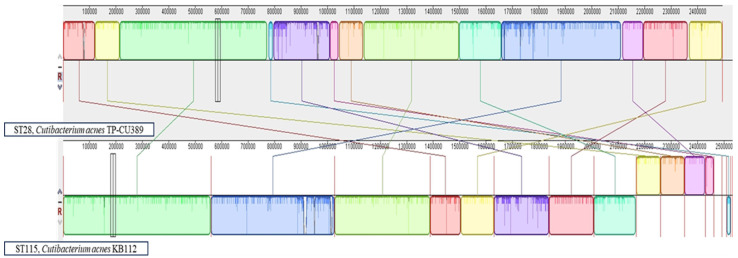
Genome comparison of *C. acnes* KB112 and *C. acnes* TPCU1 using the Mauve tool, demonstrating various rearrangements within these genomes. The internal matches are screened and classified into local collinear blocks (LCBs). LCBs show homologous areas aligned to the segment of other genomes. Similar LCBs between the genomes are connected by colored thin lines. Blocks positioned above and below the center line of the aligned area are oriented forward and backward, respectively, relative to the first genome sequence.

**Table 1 pathogens-12-01286-t001:** Homolog information, as acquired by conducting a conserved domain search.

No.	Location	Size (aa)	Description	Identity (%)	Accession No.
1	112–1194	360	DNA methyltransferase	100	WP_176453839
2	1184–2116	245	hypothetical protein (plasmid)	99.59	BBJ25236
3	2541–4460	639	TetW: tetracycline resistance ribosomal protection protein	100	WP_002586627
4	4873–5157	94	recombinase (plasmid)	100	BBJ25234
5	6269–5454	271	AAA family ATPase	100	WP_070434713
6	7687–6266	476	Tn*552* transposase (plasmid)	100	BBJ25232
7	8328–7723	202	recombinase family protein	99.5	WP_234990909
8	8931–9722	263	*erm*(50): rRNA adenine N-6-dimethyltransferase	100	WP_176453837
9	10037–10369	110	transcriptional regulator	100	WP_176453836
10	10366–11142	258	nucleotidyl transferase AbiEii/AbiGii toxin	100	WP_176453835
11	11139–11723	194	recombinase family protein	100	WP_176453834
12	12632–12979	115	plasmid mobilization relaxosome protein MobC	100	WP_176453832
13	12979–14685	569	relaxase of type IV secretion system (plasmid)	100	BBJ25226
14	14687–15277	205	DUF3801 domain-containing protein	100	WP_176453830
15	15288–16988	566	type IV secretory system conjugative DNA transfer family protein	100	WP_176453829
16	18442–18978	178	PrgI family protein	100	WP_176453825
17	19046–21316	756	ATPase of type IV secretion system (plasmid)	100	BBJ25222
18	21322–22842	506	bifunctional lytic transglycosylase/C40 family peptidase	100	WP_176453824
19	27866–28732	414	anti-repressor of ImmR (plasmid)	100	BBJ25217
20	28816–29757	323	ParA family protein	100	BBJ25216

## Data Availability

The GenBank accession numbers for the draft genome sequences of *C. acnes* KB112 and Plasmid KB112 (pKB112) are JAODIJ00000000 and OQ053204, respectively.

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
