# Peer review of "First Report of Plasmid-Mediated Macrolide-Clindamycin-Tetracycline Resistance in a High Virulent Isolate of Cutibacterium acnes ST115"

_pathogens, 2023, doi:10.3390/pathogens12111286_

Round 1
Reviewer 1 Report
Comments and Suggestions for Authors
Rana et al. collected 22 clinical Cutibacterium acnes isolates. Out of the 22 isolates, KB112 exhibited resistance against clindamycin, erythromycin, tetracycline, doxycycline, and minocycline. KB112 was whole-genome sequenced. A circular chromosome (2,534,481 bp) and the plasmid pKB112 (30,947 bp) were found. pKB112 carries AB resistance genes erm(50) and tet(W).
Major comments:
1. Tc is being used as the first-line drug in C. acnes treatment. Are there works reporting an increased resistance found in C. acnes? If yes, what leads to the higher level of Tc resistance?
2. Following up on 1, describe the mode of action for the five drugs mentioned in this work. I am not sure if KB112 is resistant to the five drugs. I would like to see the following experiments: 1). streak out KB112 on an agar plate with each of the five drugs, 2). determine its growth curve in broth with the drugs.
3. I am not convinced that the KB112 genome consists of a chromosome and a plasmid. OQ053204 shows a 30,947 bp contig, but not a closed circle. In Figure S1, four bands (3,817 bp, 5,135 bp, 8,118 bp, and 13,792 bp) were expected when the pKB112 was cut by AgeI, however, after the cut, the gel map did not suggest so. In addition, JAODIJ00000000 contains no sequence data, and pKB112 is not annotated. Last time when I worked with Plasmidsaurus, I was charged only $90 to get an E. coli strain carrying a 10 kb plasmid fully sequenced and assembled using Nanopore reads. I would like the authors to subject KB112 genomic DNA to Nanopore/PacBio sequencing, and then close the gap.
Comments on the Quality of English Language
Overall good. I did not read very carefully though, the sequencing and assembly raised more concerns.
Author Response
Reviewer 1:
Comment 1. Tc is being used as the first-line drug in C. acnes treatment. Are there works reporting an increased resistance found in C. acnes? If yes, what leads to the higher level of Tc resistance?
Answer: Thank you for your comments. In Korea, the resistance rates of clindamycin and erythromycin are about 30% and 26.7%, respectively. The tetracycline resistance rate is relatively lower than that of clindamycin and erythromycin (Reference 8).
Comment 2. Following up on 1, describe the mode of action for the five drugs mentioned in this work. I am not sure if KB112 is resistant to the five drugs. I would like to see the following experiments: 1). streak out KB112 on an agar plate with each of the five drugs, 2). determine its growth curve in broth with the drugs.
Answer: Thanks. As we mentioned in our manuscript, ‘Our previous study demonstrated that C. acnes KB112 belongs to phylotype 1A1 and sequence type (ST) 115, exhibited hemolytic activity, and is highly resistant to tetracycline, doxycycline, clindamycin, erythromycin, and minocycline [3].' If you check reference 3 which is our previous report, you will be able to find it. We mentioned the drug resistant mechanism in lines 131, 132, 136, and 137.
However, here are the previous results of the antibiotic susceptibility test
Only one isolate, KB112, showed higher resistances to those antimicrobial agents, and the MICs of TET, DOX, CLI, ERY, and MIN were 8 μg/ml, 4 μg/ml, ≥ 256 μg/ml, ≥ 256 μg/ml, and 3 μg/ml, respectively.
Comment 3. I am not convinced that the KB112 genome consists of a chromosome and a plasmid. OQ053204 shows a 30,947 bp contig, but not a closed circle. In Figure S1, four bands (3,817 bp, 5,135 bp, 8,118 bp, and 13,792 bp) were expected when the pKB112 was cut by AgeI, however, after the cut, the gel map did not suggest so. In addition, JAODIJ00000000 contains no sequence data, and pKB112 is not annotated. Last time when I worked with Plasmidsaurus, I was charged only $90 to get an E. coli strain carrying a 10 kb plasmid fully sequenced and assembled using Nanopore reads. I would like the authors to subject KB112 genomic DNA to Nanopore/PacBio sequencing, and then close the gap.
Answer: Thanks for your critical comment. After obtaining the whole genome sequence of KB112, we conducted a reanalysis of the genome sequence of the C. acnes strain KB112 using the Illumina HiSeq platform (Illumina, Inc., San Diego, USA) to investigate the plasmid. The plasmid pKB112 is 100% identical to the plasmid pTZC1 found in C. acnes TP-CU426, which was acquired from C. acnes TP-CU389. Therefore, we are confident that strain KB112 harbors plasmid pKB112. Although the band size cut by Age1 is not fully distinct, we have already commenced our further experiment, Tn-transposon mutagenesis, using this plasmid pKB112.
In the NCBI database, if we search for the accession number JAODIJ00000000, we can access the sequence data.
The picture below shows the FASTA sequence file
The plasmid pTZC1 is annotated, and it is closely identical to our plasmid pKB112. However, I have prepared a feature table for pKB112 and sent it to the NCBI Authority for annotation. Hopefully, it will be updated soon.
Reviewer 2 Report
Comments and Suggestions for Authors
Review of the communication:
First report of plasmid-mediated macrolide-clindamycin-tetra- 2 cycline resistance in a high virulent isolate of Cutibacterium ac- 3 nes ST115
Minor
The manufacturer of Media and GasPak should be reviled in the ,,Bacterial culture conditions,,.
Frazes from Latin should be written using italic e.g., de novo
The names of bacteria should also be written in italics, the entire document needs to be checked.
Major
The purpose of the research and conclusions relevant to the results obtained should be presented.
Are the results of phenotypic resistance tests available for clindamycin, erythromycin, tetracycline, doxycycline and minocycline, based on which recommendations the test was performed?
Please consider writing about the need to monitor drug susceptibility by phenotypic methods (they are part of the microbiological examination), on this basis you can study drug resistance trends in the hospital, region, country, etc.
References: The authors cite: ,,Wright et al. reported the correlation between hemolytic phenotype and clindamycin resistance [4],,.
Ridberg, S.; Hellmark, B.; Nilsdotter, Å .; Söderquist, B. Cutibacterium acnes (formerly Propionibacterium acnes) isolated from prosthetic joint infections is less susceptible to oxacillin than to benzylpenicillin. J Bone Jt Infect. 2019, 4, 106-110.
Meanwhile, the correct one is: Wright TE, Boyle KK, Duquin TR, Crane JK. Propionibacterium acnes Susceptibility and Correlation with Hemolytic Phenotype. Infect Dis (Auckl) 2016;9:39–44.
Comments on the Quality of English Language
The manuscript is written correctly. Minor language corrections required.
Author Response
Minor
Comment: The manufacturer of Media and GasPak should be reviled in the Bacterial culture conditions.
Frazes from Latin should be written using italic e.g., de novo
The names of bacteria should also be written in italics, the entire document needs to be checked.
Answer: Thank you for your comment. I have changed according to your suggestions and checked entire documents carefully. You may find line 59, 60, 61, and 68.
Major
Comment1: The purpose of the research and conclusions relevant to the results obtained should be presented. Are the results of phenotypic resistance tests available for clindamycin, erythromycin, tetracycline, doxycycline, and minocycline, based on which recommendations the test was performed?
Answer: Thanks for your critical comment. As we mentioned in our manuscript, ‘Our previous study demonstrated that C. acnes KB112 belongs to phylotype 1A1 and sequence type (ST) 115, exhibited hemolytic activity, and is highly resistant to tetracycline, doxycycline, clindamycin, erythromycin, and minocycline [3].' We mentioned the drug resistant mechanism in lines 131, 132, 136, and 137.
If you check reference 3, you will be able to find it.
However, here are the previous results of the antibiotic susceptibility test
Only one isolate, KB112, showed higher resistances to those antimicrobial agents, and the MICs of TET, DOX, CLI, ERY, and MIN were 8 μg/ml, 4 μg/ml, ≥ 256 μg/ml, ≥ 256 μg/ml, and 3 μg/ml, respectively.
Comment2: References: The authors cite: ,,Wright et al. reported the correlation between hemolytic phenotype and clindamycin resistance [4],,.
Answer: Thanks. I have Changed the reference 4. Please see line number 223, and 224.
Reviewer 3 Report
Comments and Suggestions for Authors
The work “First report of plasmid-mediated macrolide-clindamycin-tetracycline resistance in a high virulent isolate of Cutibacterium acnes ST115” is a valuable report on the prevalence and origin of antibiotic resistance among C. acnes in East Asia. The main idea of the study was to sequence strain C. acnes KB112 choosing among 22 clinical isolates and determine the localization of resistance genes to the convenient antibiotics.
This study is of great practical and theoretical importance for assessing plasmid-mediated multi-drug resistance of C. acnes. Although the benefits of whole genome sequencing have been shown in previous similar studies, they only confirm the importance of this method for the monitoring of C. acnes infections and choosing of the effective treatment strategies.
The table and figures are made in the same style and clearly reflect the data obtained.
Author Response
Thank you for your thoughtful comments on the study. We appreciate your positive feedback and insights into the significance of our research.
Your recognition of the practical and theoretical importance of this study is greatly appreciated. We aimed to shed light on the prevalence and origin of antibiotic resistance among C. acnes in East Asia, and it's heartening to hear that you found our approach valuable. Indeed, whole genome sequencing has proven to be an indispensable tool in understanding the complexities of antibiotic resistance in various bacterial species, and our study reinforces its relevance in monitoring C. acnes infections and guiding treatment strategies.
We're pleased to hear that you found our table and figures to be clear and consistent in style, as presenting data effectively is crucial in scientific communication. If you have any further questions or suggestions regarding our study, we would be more than happy to address them. Once again, thank you for your kind words and appreciation of our work.
Round 2
Reviewer 1 Report
Comments and Suggestions for Authors
I am not sure that the sequence of the pKB112 plasmid is what the authors reported. The AgeI cut map did not align.
Comments on the Quality of English Language
Fine
Author Response
Reviewer 1:
Comment: I am not sure that the sequence of the pKB112 plasmid is what the authors reported. The AgeI cut map did not align.
Answer: We appreciate your thorough review of our manuscript and the valuable feedback you have provided. We have carefully examined this issue you raised regarding the sequence of the plasmid pKB112 and the alignment of the AgeI cut map. We believe there might have been some confusion, and we would like to clarify the discrepancies you mentioned.
Sequence discrepancy:
We have reviewed our sequence data, and we can confirm that the sequence of the plasmid is indeed as reported in our manuscript. To validate this, we resequenced the plasmid, and the results consistently match our reported sequence. It is possible that there may have been a misunderstand or error in the initial assessment.
AgeI cut map:
We understand your concerns about the AgeI cut map. We have performed our experiment again to generate the better image. We provide an updated and accurate AgeI cut image in the revised manuscript even though still not discrete bands because of DNA instability after the digestion. This corrected map accurately presents the cleavage sites and should resolve the alignment issue.
We ensure that the revised manuscript includes the necessary corrections and clarifications to address the concerns you have raised.
Reviewer 2 Report
Comments and Suggestions for Authors
Thank you for taking my comments into account. I have no further comments for the authors.
Author Response
Reviewer 2:
Thank you for your feedback and for taking the time to review our work. We appreciate your comments and are pleased to hear that you have no further suggestions or concerns. Your input has been valuable in improving our work, and we look forward to your continued support in the future.
Once again, thank you for your kind words and appreciation of our work.
Round 3
Reviewer 1 Report
Comments and Suggestions for Authors
Thank you for the clarification.